# Human visual grouping based on within- and cross-area temporal correlations

**Yen-Ju Chen***, **Zitang Sun, Shin'ya Nishida**

Graduate School of Informatics, Kyoto University, Kyoto, Japan

* chen.yen-ju.t05@kyoto-u.jp

## Abstract

Perceptual organization in the human visual system involves neural mechanisms that spatially group and segment image areas based on local feature similarities, such as the temporal correlation of luminance changes. Successful segmentation models in computer vision, including graph-based algorithms and vision transformer, leverage similarity computations across all elements in an image, suggest that effective similarity-based grouping should rely on a global computational process. However, whether human vision employs a similarly global computation remains unclear due to the absence of appropriate methods for manipulating similarity matrices across multiple elements within a stimulus. To investigate how "temporal similarity structures" influence human visual segmentation, we developed a stimulus generation algorithm based on Vision Transformer. This algorithm independently controls within-area and cross-area similarities by adjusting the temporal correlation of luminance, color, and spatial phase attributes. To assess human segmentation performance with these generated texture stimuli, participants completed a temporal two-alternative forced-choice task, identifying which of two intervals contained a segmentable texture. The results showed that segmentation performance is significantly influenced by the configuration of both within- and cross-correlation across the elements, regardless of attribute type. Furthermore, human performance is closely aligned with predictions from a graph-based computational model, suggesting that human texture segmentation can be approximated by a global computational process that optimally integrates pairwise similarities across multiple elements.

### Author Summary

How does the human visual system use temporal information to segment objects in a dynamic scene? When observing ever-changing environments, our brains must determine which regions belong to the same object and which are distinct. However, the mechanisms underlying this process remain poorly understood. In this study, we investigate how "temporal similarity structures"—patterns of

**Data availability statement:** All resources, including the experiment code, demo video, raw data, high-resolution figures in the paper, and bootstrapping results, have been uploaded to the Open Science Framework platform. The project can be accessed at the following link: https://osf.io/9yncd/?view_only=eb481f-2b28ab42e29ff267833f70b8e5.

**Funding:** This work was supported in part by the Spring Fellowship (Grant Number JPMJFS2123) for SZ and YJC and in part by JSPS Grants-in-Aid for Scientific Research (KAKENHI) (Grant Numbers JP20H00603, JP20H05950, JP20H05957, and 24H00721) for SN. The funders had no role in study design, data collection and analysis, decision to publish, or preparation of the manuscript.

**Competing interests:** The authors have declared that no competing interests exist.

correlation over time—affect visual segmentation. We developed a novel method for generating dynamic stimuli with precisely controlled temporal similarity and systematically tested how within-area and cross-area temporal correlations influence segmentation. Participants performed a task in which they identified segmentable textures, and the results showed that segmentation performance improves when regions exhibit strong internal consistency but lower similarity with adjacent regions. Our findings revealed that human visual segmentation relies on a global computational mechanism that integrates temporal similarity cues to distinguish visual structures. Additionally, our stimulus generation framework provides a powerful tool for future research on perceptual organization and mid-level vision.

## Introduction

Visual perceptual organization is a fundamental function of the human visual system that enables the grouping of objects based on various attributes, such as luminance, color, and temporal dynamics. These attributes are captured locally by different neurons, each of which has a limited receptive field. To segment a coherent object from its surroundings, an additional process is required to compare local neuronal signals across space and determine whether they belong to the same object. One of the key computational principles underlying this process is similarity-based grouping, a concept introduced in Gestalt psychology, which posits that elements with similar features tend to be grouped together [1,2]. Similarity operates across a wide range of visual attributes, both spatial and temporal. Spatial cues include orientation, color hue [3–5], and luminance [6], as well as motion properties such as direction and speed (i.e., common fate) [7,8]. Temporal cues involve temporal synchrony [9–11], temporal structure [12–14], and generalized common fate [15]. The present study investigated the computational mechanisms of similarity-based grouping, focusing on the temporal correlation of local feature changes. Although our argument is primarily based on temporal cues, the proposed framework may be generalized to similarity-based grouping involving other visual attributes.

From a computational perspective, the decision to group two local elements depends not only on their pairwise similarity but also on the similarities between all other element pairs in the visual scene. For example, consider a spatial arrangement of four elements (A, B, C, and D), as illustrated in Fig 1. If the pairwise similarity is high within {A, B} and within {C, D}, but relatively low for all other element pairs, the most computationally reasonable solution is to segregate the elements into two distinct groups: {A, B} and {C, D}. To achieve this, the visual system must first evaluate the pairwise similarities between all local elements and then derive a global solution that maximizes within-group similarities while minimizing cross-group similarities. When the number of elements is small (Fig 1A), this computation is relatively straightforward. However, as the number of elements increases, the complexity of the grouping task grows exponentially (Fig 1B).

(A) Stimulus — Perception — Neural analysis

Physical similarity matrix

|   | A | B | C | D |
|---|---|---|---|---|
| A | 1.00 | 0.99 | 0.61 | 0.53 |
| B | 0.99 | 1.00 | 0.70 | 0.61 |
| C | 0.61 | 0.70 | 1.00 | 0.99 |
| D | 0.53 | 0.61 | 0.99 | 1.00 |

Perceptual connection matrix

|   | A | B | C | D |
|---|---|---|---|---|
| A | 1.00 | 1.00 | 0.00 | 0.00 |
| B | 1.00 | 1.00 | 0.00 | 0.00 |
| C | 0.00 | 0.00 | 1.00 | 1.00 |
| D | 0.00 | 0.00 | 1.00 | 1.00 |

(B) Rapid growth of complexity with the number of elements

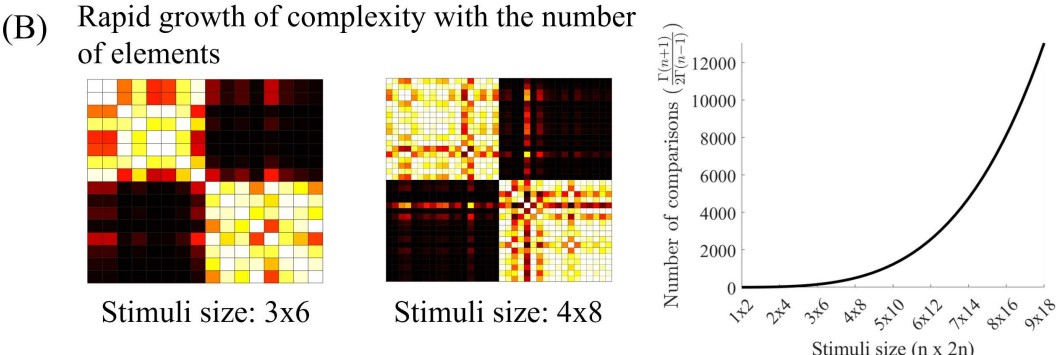

Stimuli size: 3x6          Stimuli size: 4x8

Number of comparisons $\left(\frac{\Gamma(n+1)}{2\Gamma(n-1)}\right)$ vs Stimuli size (n x 2n)

**Fig 1. (A)** Illustration of grouping and segmentation based on the global similarity structure when the number of elements is small. A, B, C, and D represent stimulus elements at different spatial locations. The physical pairwise similarities are assumed to be high between A and B and between C and D, while they are relatively low for the other four pairs, as indicated by the physical similarity matrix. This configuration potentially leads to the grouping of {A, B} and {C, D}, as depicted by the dashed red squares. Following neural processing that incorporates the global similarity structure, similar pairs are grouped, whereas dissimilar pairs are segregated, as reflected in the perceptual connection matrix. **(B)** Demonstration of how computational complexity increases with stimulus size. The left panel presents the physical similarity matrix for 3×6 and 4×8 element stimuli, respectively. The right panel illustrates the exponential growth in the number of pairwise comparisons as the number of elements increases.

Various global computational approaches have been proposed in the field of computer vision (CV) to address visual segmentation. Self-attention transformers, which leverage pairwise similarity across all elements, have demonstrated performance comparable to human grouping abilities [16]. Similarly, Conditional Random Fields (CRFs), a specialized form of Markov Random Fields (MRFs), have been successfully applied to model dependencies between local similarities, enabling high-quality image segmentation [17]. Additionally, graph-based methods, such as eigendecomposition on the graph Laplacian—a linear transformation of the similarity matrix—have achieved state-of-the-art performance [18,19].

These approaches share similarities with human perceptual strategies, particularly in motion-based segmentation tasks [20–22].

Despite these advancements, previous studies on human vision have not systematically examined how the global similarity structure of a stimulus influences perceptual grouping. Prior research has primarily manipulated either within-group similarity or cross-group similarity, but not both simultaneously. For example, studies by Lee and Blake (1999) and Morgan and Castet (2002) modulated the level of temporal correlation in motion direction changes within a target area, thereby altering within-group similarity while leaving cross-group similarity uncontrolled [13,23]. Similarly, other studies have focused on modulating within-group similarity while keeping cross-group similarity close to zero [11,15,24].

One possible reason for the lack of research on the effects of global similarity structure in human perceptual grouping is the technical challenge of independently manipulating different aspects of similarity. Specifically, modifying the similarity of a single element inevitably affects its relationship with all other elements in the scene. To overcome this limitation, we developed a novel stimulus generation method that leverages a vision transformer (ViT) [25]. This approach allowed us to precisely manipulate pairwise similarities across all element pairs. Our stimulus design focused on the temporal effects of visual segmentation, defining similarity between element pairs based on the correlation of their temporal sequences. By utilizing well-defined loss functions, our stimuli were free from spatial artifacts, enabling us to isolate and examine human performance in purely similarity-based visual segmentation tasks.

This study makes three key contributions. First, we introduce and evaluate a ViT-based stimulus generator with a customized loss function that enables precise control over pairwise element temporal similarity. Second, using the generated stimuli, we conduct a psychophysical experiment to examine how perceptual segregation is influenced by both within-group and cross-group temporal similarities. Finally, we introduce a graph cut model—partially based on previous work [18]—as a simple segmentation framework to assess whether computations based on the global "temporal similarity structure" can approximate human perceptual performance.

## Study 1. Stimulus generation incorporating ViT-based generator

### Methodology

**ViT-generated feature tensor.** This section describes our approach to generating a 2D spatiotemporal stimulus using a lightweight ViT. The goal of the model was to generate a feature tensor $F^* \in \mathbb{R}^{H \times W \times N}$ from a sampled noise tensor $F \in \mathbb{R}^{H \times W \times N}$, satisfying user-assigned cosine similarity (temporal correlation) constraints in two within-areas, denote as ($\xi_A$ or $\xi_B$) and across two areas, denote as ($\xi_{Cross}$) (Fig 2).

1. Problem definition

- We use a 2D feature map to represent spatiotemporal data. Consider a feature tensor $F$, defined on a spatial grid $G$, where $H$, $W \in \mathbb{N}^+$ represent the height and width of the spatial grid. The spatial grid is defined as $G = \{(i,j) | 1 \leq i \leq H, \ 1 \leq j \leq W\}$. At each spatial location $(i, j)$, there exists a corresponding temporal sequence $P_{ij} \in \mathbb{R}^N$. Here, N represents the length of the temporal sequence, set to 30 frames. $F$ can be formally defined as follows:

$$F : G \rightarrow \mathbb{R}^N, \ F(i,j) = P_{ij} \in \mathbb{R}^N, \ s.t. \ \forall (i,j) \in G \tag{1}$$

• Splitting into sub-areas: To impose specific temporal correlation constraints, we partitioned $G$ into two subsets, $\varphi_A$ and $\varphi_B$:

$$\phi_A = \left\{ (i, j) \middle| 1 \leq i \leq H, \ 1 \leq j \leq \frac{W}{2} \right\}, \ \phi_B = \left\{ (i, j) \middle| 1 \leq i \leq H, \ \frac{W}{2} < j \leq W \right\} \tag{2}$$

Here, we set H as 8 and W as 16. Therefore, the feature tensor is an $8 \times 16$ rectangle, with two $8 \times 8$ square sub-areas on the left and right sides.

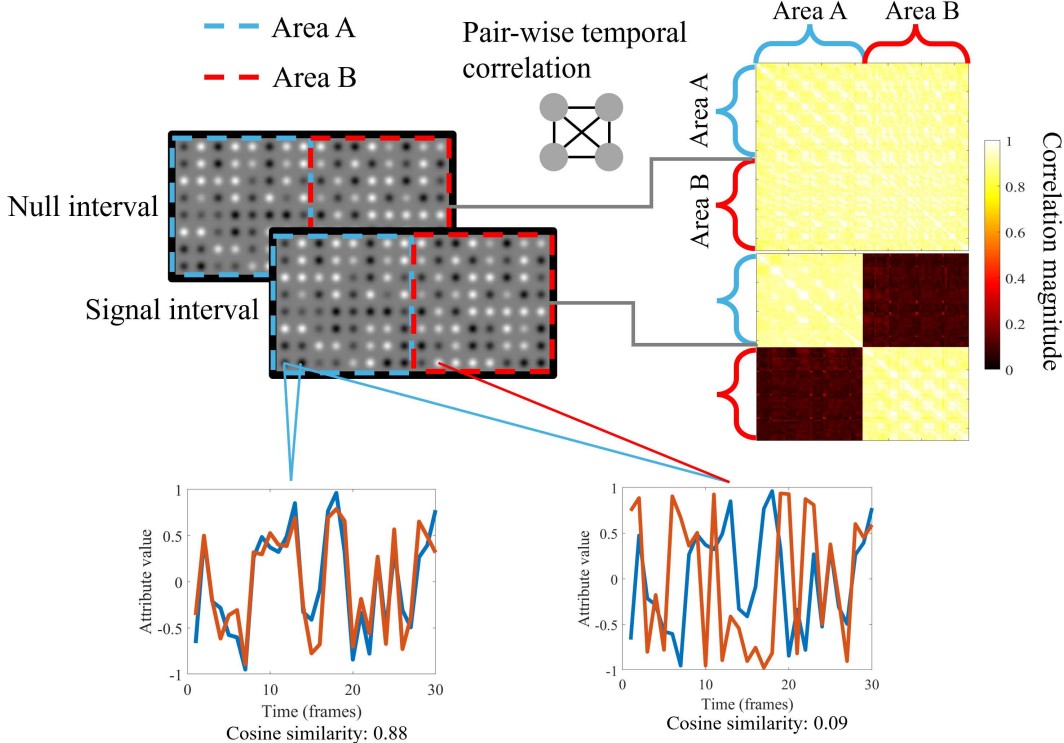

**Fig 2. Illustration of the generated experimental stimuli.** The luminance bulb is shown as an example here, but in the actual experiment, the stimulus could be a luminance bulb, color bulb, or luminance Gabor. The pairwise similarity heatmap is shown on the right; the space is flattened into a single dimension for convenient plotting. The position on the matrix corresponding to the 2D location is labeled, with the blue part corresponding to Area A and the red part corresponding to Area **B**. A demo time series is plotted at the bottom; the left figure demonstrates the elements with high temporal similarity, while the right figure demonstrates those with low similarity. The demo video can be viewed at https://osf.io/9yncd/?view_only=eb481f2b28ab42e29ff267833f70b8e5.

## 2. Temporal correlation constraints

We aimed to generate $F^*$, whose spatiotemporal patches exhibit assigned cosine similarities. The similarities were calculated using the following formula:

$$\text{sim}\left(\mathsf{P}_{ij}, \mathsf{P}_{i'j'}\right) = \frac{\mathsf{P}_{ij}^{\top} \mathsf{P}_{i'j'}}{\|\mathsf{P}_{ij}\|_2 \|\mathsf{P}_{i'j'}\|_2} \tag{3}$$

The similarity was calculated between two vectors at spatial locations $(i, j)$ and $(i', j')$. We intended to add the following constrains:

• Within-area temporal correlation: For any pair $((i, j), (i', j'))$ where both $(i, j)$ *and* $(i', j')$ belong to the same area ($\phi_A$ or $\phi_B$), the similarity should be close to a desired constant, i.e., $\xi_A$ and $\xi_B$, respectively. This can be expressed as follows:

$$\text{sim}\left(\mathbf{P}_{ij}, \mathbf{P}_{i'j'}\right) \approx \xi_A, \ \forall (i,j), (i',j') \in \phi_A,$$
$$\text{sim}\left(\mathbf{P}_{ij}, \mathbf{P}_{i'j'}\right) \approx \xi_B, \ \forall (i,j), (i',j') \in \phi_B \tag{4}$$

Throughout this study, $\xi_A$ and $\xi_B$ were kept constant.

- Cross-area temporal correlation: For any pair $((i, j), (i', j'))$ spanning different sub-areas, i.e., $(i,j) \in \phi_A$ and $(i', j') \in \phi_B$ (or vice versa), the similarity should be close to a distinct constant $\xi_{cross}$:

$$\mathrm{sim}\left(\mathbf{P}_{ij}, \mathbf{P}_{i'j'}\right) \approx \xi_{cross}, \ \forall (i,j) \in \phi_A, \ (i',j') \in \phi_B \tag{5}$$

In addition to enforcing temporal correlation constraints, we aim to remove any other distinguishing features that could facilitate segmentation between $\varphi_A$ and $\varphi_B$, such as spatial cues (e.g., luminance or contrast differences), texture patterns, or higher-order statistical irregularities. In other words, our goal is to ensure that the stimuli rely solely on temporal correlation differences for segmentation. Ideally, these stimuli would exhibit i.i.d. white noise–like temporal spectra, subject only to the specified temporal correlation constraints. Finding an analytical solution for $F^*$ that satisfies all these constraints could be mathematically challenging. Instead, we leveraged a neural generative approach using a lightweight ViT model to get the numerical approximation.

3. Transformer-based generative model

Our generative model was based on a ViT backbone, incorporating convolutional residual blocks at both the encoding and decoding stages. The architecture can be summarized as follows:

- Encoder: A series of 2D CNN residual blocks [26] processed an initial latent tensor $F$, expanding the feature dimension from N to 128. In our implementation, we used four-stacked *ResidualBlock2D* modules, each consisting of convolution, normalization (InstanceNorm2D), and *GELU* activation, with the residual connection between two layers.

- Transformer layers: We flattened the spatial dimensions and treated each location as a "token" in standard transformer terminology. Each token is represented by 128 dimension features, attended to every other token via multi-head self-attention (a similarity representation based on dot-product distance). Stacking multiple such layers enabled the model to capture global temporal correlations across the entire feature map.

- Decoder: Another three layers of convolutional layers decode the features into the target spatiotemporal volume $F^*$, condense the feature dim from 128 to N. We applied non-linear activations (tanh) to ensure an output within a controlled numeric range (-1,1).

Mathematically, we defined the transformer layers as a function $\mathcal{T}(\cdot)$. The overall generation process transform a white-noise prior into target stimuli, could be expressed as:

$$F^* = \mathrm{Dec}\left(\mathcal{T}\left(\mathrm{Enc}(F)\right)\right) \tag{6}$$

The model parameters were optimized so that the resulting output $F^*$ fulfilled the cosine similarity constraints defined previously. The transformer was selected because the self-attention mechanism in transformers computes pairwise temporal correlations (via the Query, Key, and Value operations) between all tokens in a layer. This is precisely the structural property we intended to exploit, aiming to modulate temporal correlations (cosine similarity) within and across spatial locations in $\phi_A$ and $\phi_B$. By stacking multiple transformer layers, the model could iteratively adjust and refine these pairwise relationships, converging on a solution that respects the target within-area and cross-area similarities.

4. Optimization

We optimized the model parameters to produce a stimulus $F^*$, which enforced both the within-area (within $\phi_A$ and $\phi_B$) and cross-area temporal correlations described earlier while also constraining the signal to have similar mean and variance values across areas.

The input for the generative model was initialized as a white noise tensor, i.e., random values drawn from a normal distribution with zero mean and moderate variance, encouraging broad coverage of frequencies.

To optimize the stimulus $F^*$, we defined a customized loss function $\mathcal{L}_{total}$ composed of several terms, each constrain a different statistical property:

- **Within-area temporal correlation constraints:** We computed the cosine similarity among all pairs of vectors within each sub-area. This resulted in two similarity matrices, $S^A, S^B \in \mathbb{R}^{(H \times \frac{W}{2}) \times (H \times \frac{W}{2})}$, where $S^A$ and $S^B$ represent similarity values within the regions $\phi_A$ and $\phi_B$, respectively. The within-area loss is defined as the summation of the absolute differences between the computed similarities and the desired similarity constants $\xi_A$ and $\xi_B$. The within-area temporal correlation loss $\mathcal{L}_A$ and $\mathcal{L}_B$ were then:

$$\mathcal{L}_A = \left\| \left| \mathbf{S}^A \right| - \xi_A \right\|_1, \ \mathcal{L}_B = \left\| \left| \mathbf{S}^B \right| - \xi_B \right\|_1 \tag{7}$$

- **Cross-area temporal correlation constraints:** The cosine similarity tensor $S^{AB} \in \mathbb{R}^{H \times \frac{W}{2} \times H \times \frac{W}{2}}$ was computed between the space spanning the areas $\phi_A$ and $\phi_B$. The cross-area temporal correlation loss $\mathcal{L}_{AB}$ was then:

$$\mathcal{L}_{AB} = \left\| \left| \mathbf{S}^{AB} \right| - \xi_{cross} \right\|_1. \tag{8}$$

- **Mean and variance regularization:** To maintain the whole tensor following a similar feature distribution, we added a loss to let the generated stimuli converge on specific statistics. Let $F^A, F^B \subset F$ denote the subsets of F corresponding to $\varphi_A$ and $\varphi_B$, respectively. We aimed to match both global mean ($\mu_G$) and variance ($\sigma_G^2$) values, defined as:

$$\begin{aligned} \mathcal{L}_{D,A} &= \left| \text{Var}\left(F^A\right) - \sigma_G^2 \right| + \left| \text{Mean}\left(\mathbf{F}^A\right) - \mu_G \right| \\ \mathcal{L}_{D,B} &= \left| \text{Var}\left(\mathbf{F}^B\right) - \sigma_G^2 \right| + \left| \text{Mean}\left(\mathbf{F}^B\right) - \mu_G \right|. \end{aligned} \tag{9}$$

$Var(\cdot)$ and $Mean(\cdot)$ computed a single scalar for variance and the mean for the whole input tensor, respectively. Here, we set $\mu_G$ as 0 and $\sigma_G^2$ as $\frac{1}{3}$ to approximate a uniform distribution between −1 and 1.

- **Equalizing global statistics:** To ensure that the feature distributions across $\varphi_A$ and $\varphi_B$ exhibit similar mean and variance statistics, we enforced constraints on the mean and variance of each sub-region. We defined an additional penalty:

$$\mathcal{L}_{D,AB} = \left| Mean\left(F^A\right) - Mean\left(F^B\right) \right| + \left| Var\left(F^A\right) - Var\left(F^B\right) \right| \tag{10}$$

- **Variance expansion ("white noise" tendency):** To increase variability within each sub-area and push the solution towards a white noise-like distribution, we applied a negative penalty on intra-area pairwise distances:

$$\begin{aligned} \mathcal{L}_{V,A} &= -Mean \left\| F^A_{(i,j,:)} - F^A_{(i,j,:)} \right\|_1 \\ \mathcal{L}_{V,B} &== -Mean \left\| F^B_{(i,j,:)} - F^B_{(i,j,:)} \right\|_1 \end{aligned} \tag{11}$$

By negatively weighting the average pairwise differences, we encouraged the model to spread out feature values, thereby increasing variance.

- **Final loss function:** Combining all of the above components, we obtained the total loss as follows:

$$\mathcal{L}_{total} = \alpha_1 \left(\mathcal{L}_A + \mathcal{L}_B\right) + \alpha_2 \mathcal{L}_{AB} + \alpha_3 \left(\mathcal{L}_{D,A} + \mathcal{L}_{D,B}\right) + \alpha_4 \mathcal{L}_{D,AB} + \alpha_5 \left(\mathcal{L}_{V,A} + \mathcal{L}_{V,B}\right) \tag{12}$$

where $\alpha_k$ are scalar weights to balance the contributions of each term.

5. Training procedure

The network parameters (including those of the transformer and the encoder/decoder blocks) were optimized jointly via an AdamW optimizer. At each iteration, we:

- Generated an output stimulus $F^*$ from the current model.

- Computed all required similarity, variance, and mean statistics to obtain $\mathcal{L}_{total}$.

- Performed backpropagation to update model parameters.

Since we began with white noise initialization and imposed temporal correlation constraints alongside variance/mean regularizations, the final output remained statistically rich (i.e., maintained high variability) while respecting the assigned within- and cross-area temporal correlation targets. This approach produced a refined $F^*$ that approximated the ideal solution to our temporal correlation objectives. We generated 30 different $F^*$ values for each temporal correlation pair.

## Results

### Invisibility of spatial cues

To confirm that the generated stimuli could only be distinguished based on the designated temporal similarity structure, we first verified that they were unlikely to be differentiated by other spatial indicators. We calculated two standard measures of spatial cues: the spatial average value per frame, representing the attribute's convergent value (e.g., mean luminance), and the spatial root mean square deviation per frame, indicating the attribute's concentration magnitude (e.g., luminance contrast). These statistics were selected based on previous studies demonstrating that the human visual system is particularly sensitive to such regularities when evaluating texture [6,27].

Analysis of the first index revealed that the mean luminance difference between the left and right partitions (Areas A and B in Fig 2) across 30 frames was 0.17% for all temporal correlation pairs, with a peak difference of 0.24%. This negligible discrepancy suggested that participants were unlikely to perceive a difference based on mean luminance alone. Similarly, for the second index, the findings indicated that the average root mean square deviation difference between partitions across 30 frames was 0.34% for all temporal correlation pairs, reaching a maximum of 0.61%. This magnitude was well below the threshold required for human perception of such differences, confirming that spatial cues were effectively minimized.

### Validation of cosine similarity distribution

Next, we assessed whether the stimuli generated by the ViT exhibited the designated cosine similarity values. presents the temporal correlation distribution, obtained by directly computing the absolute cosine similarity of the generated stimuli. The Fig 3 demonstrates that both within-area and cross-area temporal correlations closely align with the assigned values. Furthermore, additional quantitative analysis revealed high predictability of the measured temporal correlation based on the assigned values ($R^2 = 0.99$). These results demonstrated that our generation protocol effectively manipulated the similarity structure of the stimuli as intended.

### Quasi-white noise characteristics

Finally, we analyzed the spectral properties of the generated stimuli. The temporal spectrum for Areas A and B was computed separately (Fig 4). Regardless of the temporal correlation pair, the spectrum exhibited a near-uniform distribution, indicating that the generated stimuli closely resemble white noise. This broadband energy distribution ensures activation of diverse neural mechanisms. Additionally, there were no significant spectral differences between Areas A and B during the signal interval, suggesting that simple linear operations would be insufficient to distinguish between the two partitions.

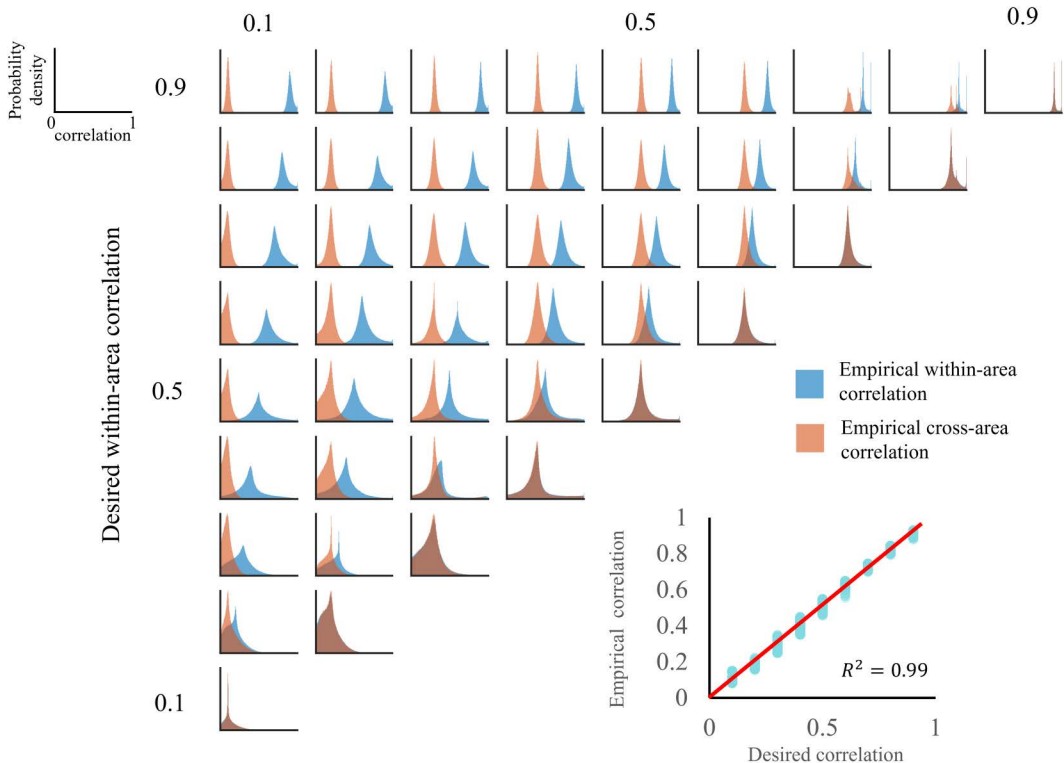

**Fig 3. Distribution of temporal correlations in the generated stimuli.** The assigned within- and cross-area correlation values are shown on the axis labels. Each subgraph displays the empirical temporal correlation distribution separately for within- and cross-area temporal correlations, with within-area temporal correlations represented in blue and cross-area temporal correlations in orange. The lower-right quadrant presents a scatter plot of the desired temporal correlation values (x-axis) versus the actual values (y-axis). The red line represents the identity line (y = x) as a reference.

## Exclusion of higher-order visual feature visibility

Our stimulus design also excluded higher-order statistical features that could be used in the segmentation task, such as local motion differences and temporal pattern recognition.

Humans can perceive shape based on local motion differences (i.e., shape-from-motion) [8,28,29]. However, our stimulus design mitigates this possibility. Temporal similarity was computed using the absolute value of cosine similarity, meaning that simultaneous increases or decreases in brightness were treated as having the same magnitude of temporal similarity. This approach ensured that any potential motion signals were drift-balanced across the left and right partitions, eliminating systematic directional motion cues that could facilitate segmentation.

Additionally, humans can distinguish regions by recognizing different temporal regularities after long exposure or repeated presentations [30]. However, our stimuli were designed to approximate quasi-white noise in the temporal domain, making it difficult to extract stable, discernible patterns in a single exposure. Given the short presentation duration and the absence of repeated trials with the same sequence, participants likely lacked sufficient information to develop reliable expectations about temporal structures.

## Study 2. Human segmentation performance for the generated stimuli

In this section, we report the results of a psychophysical experiment designed to investigate how within- and cross-area temporal correlations influence human segmentation performance. By independently manipulating within- and cross-area

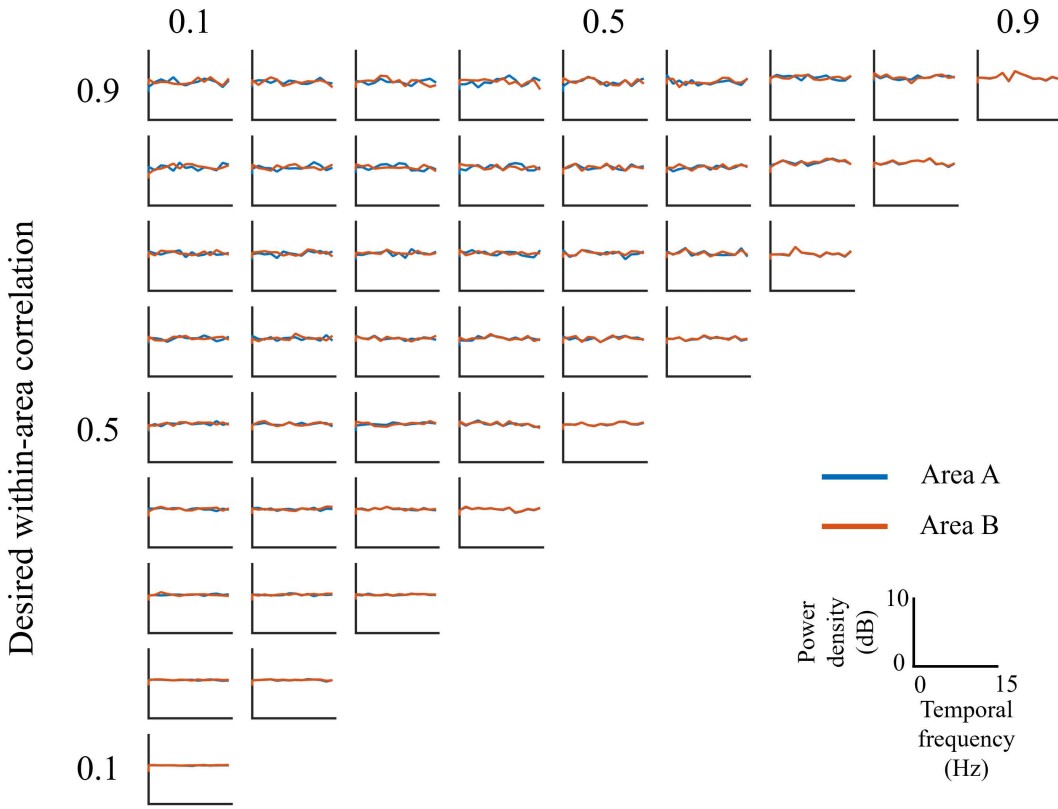

**Fig 4. Temporal spectrum of the generated stimuli.** The assigned within- and cross-area temporal correlation values are indicated on the axis labels. The mean power spectrum within Area A and Area B is shown in blue and orange, respectively. All subgraphs share the same axis scaling, as depicted in the lower-right section of the figure, where the x-axis represents temporal frequency (Hz) and the y-axis represents power density (dB).

correlations using the stimulus generation method described in the last section, we directly test how human reperformance covaries with structured changes in the similarity matrix. The results are expected to provide insights into whether segmentation depends on simple local statistics or on a global computation that integrates pairwise correlations across spatial locations. The results indicate that both within- and cross-area temporal correlations influence human segmentation performance in an antagonistic way. In the next section (Study 3), we will computationally analyze the pattern of the obtained results using a model of texture segregation based on a graph-cut algorithm.

## Methodology

### Ethics statement

The study protocol was conducted in accordance with the ethical standards of the Declaration of Helsinki and was approved by the Ethics Committee of Kyoto University (KUIS-EAR-2020–003). All participants received full explanations of the experimental procedures and provided written informed consent before the experiment began.

### Apparatus

The stimuli were displayed on a VIEWPixx/3D LCD monitor (VPixx Technologies, Saint-Bruno-de-Montarville, Canada) with a resolution of 1,920 × 1,080 pixels and a refresh rate of 30 Hz. The lowest, mean, and highest display luminance

ranged from 1.8 cd/m² (lowest) to 96.7 cd/m² (highest), with a mean luminance of 48.4 cd/m². The linear output of luminance was calibrated for each channel using an i1Pro chromometer (VPixx Technologies). Each pixel subtended 1.3 arcmin at a viewing distance of 70 cm. Participants were seated in a dark room with a chinrest to stabilize head position. The experiments were programmed using the Psychophysics Toolbox [31] in MATLAB release 2022a (MathWorks, Natick, MA, USA). The study protocol adhered to the ethical standards of the Declaration of Helsinki, except for preregistration, and was approved by the Ethics Committee of Kyoto University (KUIS-EAR-2020–003).

## Stimuli

The actual stimulus (demonstrated in Fig 2) was based on the generated sequence $F^*_{8 \times 16 \times 30}$, randomly drawn from the 30 variants. The stimulus sequence for one presentation was a dynamic 8 × 16 element lattice lasting for 30 frames (1 s). Each element subtended 0.5° in diameter, with the entire lattice subtending 4° × 8°. Each value in $F^*$ was transformed into a 0.5° × 0.5° element within the lattice, and all elements were concatenated, yielding the final stimulus. With luminance as the attribute, the element was defined as a Gaussian bulb as follows:

$$L(x, y, t) = L_{mean} \cdot \left(1 + F^*(i, j, t) \cdot e^{\frac{x^2+y^2}{\sigma^2}}\right)$$

(13)

Where x and y are spatial locations on the plane, 0 is the center of the element, and t is the frame. i, j is the location within $F^*$, where i is an integer ranging from 1 to 8, and j is an integer ranging from 1 to 16. $L_{mean}$ is the mean luminance of the screen, and $\sigma$ is the Gaussian windows of the element, set as 5.25 arcmin.

The element was also a Gaussian bulb with color as the attribute, but with three separate channel assignments:

$$L(x, y, t) = \begin{cases} R(x, y, t) = L_{mean} \cdot \left(1 + F^*(i, j, t) \cdot e^{\frac{x^2+y^2}{\sigma^2}}\right) \\ G(x, y, t) = L_{Iso.RG} \cdot L_{mean} \cdot \left(1 - F^*(i, j, t) \cdot e^{\frac{x^2+y^2}{\sigma^2}}\right) \\ B(x, y, t) = L_{mean} \end{cases}$$

(14)

where $L_{Iso.RG}$ is the iso-luminant index between red and green.

Taking spatial phase as the attribute, the element changed to the Gabor patch as follows:

$$L(x, y, t) = L_{mean} \cdot \left(1 + \cos\left(2\pi\omega r - \frac{\pi}{4} F^*(i, j, t)\right) \cdot e^{\frac{x^2+y^2}{\sigma^2}}\right)$$
$$r = y\cos(\theta) + x\sin(\theta)$$

(15)

where $\omega$ is the spatial frequency, set to 2 cpd, and $\theta$ is the orientation of the Gabor pattern, which was randomly drawn from the uniform distribution amongst $[0, 2\pi)$. The $\frac{\pi}{4}$ term in advance the $F^*$ ensured that the spatial phase change could not exceed 90° per frame, thereby preventing any effect of luminance flickering.

## Participants

The experiment included two of the authors and three naïve participants (five males; mean age: 25.8 years), all of whom reported normal or corrected-to-normal vision. Participants were informed of the study's nature and provided written informed consent before participation. Monetary compensation was provided upon completion. Prior to the main experiment, a revised version of the minimum motion method ([32,33] was used to measure the equiluminant ratio between the red (R) and green (G) channels for each participant. Participants viewed a rightward-drifting Gabor, where the R and G channels were 180° out-of-phase, and adjusted the channel ratio to minimize the perception of motion. The mean R:G ratio was determined to be 0.58.

## Design and procedure

To measure participants' segmentation performance, we adopted a two-interval forced choice (2IFC) paradigm. In each trial, participants sequentially viewed two 1,000-ms video clips, separated by a 100-ms interval. One clip, designated as the signal interval, was distinguishable due to a difference between within- and cross-area temporal correlations. The other clip, referred to as the null interval, had matched within- and cross-area correlations, with the same value as the signal interval's within-area correlation, and thus contained no segmentation cue by design. An example is shown in Fig 2. Participants were instructed to identify the interval in which the texture appeared divided into two distinct regions and indicated their choice using a keyboard.

We tested nine levels of cosine similarity (0.1 to 0.9, in steps of 0.1) for both within- and cross-area correlations. Instead of testing all 81 possible pairs, we tested 45 pairs in which the within-area correlation was greater than or equal to the cross-area correlation. Pairs where the cross-area correlation exceeded the within-area correlation were excluded, as no segmentation was expected under such conditions. For each correlation pair, participants completed 30 trials of the 2IFC task.

To examine the generalizability of temporal correlation–based segmentation, we included three visual attributes: luminance, color (red/green), and spatial phase. Each participant completed all 3 attributes × 45 correlation pairs × 30 trials, totaling 4,050 trials. The three attribute conditions were tested sequentially, and their order was randomized across participants.

To minimize fatigue, each attribute condition was divided into 3 blocks, with each block containing 10 trials for each of the 45 correlation pairs (450 trials per block). Each block lasted approximately 50 minutes, and participants could rest as long as needed between blocks. In total, the experiment consisted of 9 blocks and took approximately 7.5 hours to complete, excluding break time.

## Results

Fig 5 illustrates the participants' segmentation accuracy, expressed as the proportion of correct responses, across different temporal correlation pairs and for the three visual attributes: luminance, color, and spatial phase. We also show the contour plot version in Fig 6 to show the general tendency. A consistent trend was observed across all attributes, with segmentation performance varying as a function of both cross-area and within-area temporal correlations. Specifically, segmentation perception improved as the within-area temporal correlation increased and the cross-area temporal correlation decreased.

The results indicated that visual segmentation performance improves only when the difference between within- and cross-area temporal correlations is sufficiently large. Although the general trend remained consistent across attributes, we observed variations in overall accuracy, with luminance yielding the highest accuracy, followed by color and then spatial phase. This suggested that segmentation based on temporal similarity structure is more challenging when the distinguishing attribute is spatial phase compared to luminance or color. Additionally, the contour of segmentation performance in Fig 5 deviated slightly from parallel alignment with the diagonal line (where the within-area temporal correlation equals the cross-area temporal correlation). This suggested that within-area and cross-area temporal correlations do not symmetrically influence segmentation performance. A possible explanation for this phenomenon may lie in the spatial arrangement of the cross-area pairs, where the mean spatial distance is twice that of the within-area pairs.

## Study 3. Analysis of human performance using a graph cut model

The results presented in the previous section have indicated that human visual processing incorporates both within- and cross-area temporal correlations in similarity-based grouping, demonstrating its ability to process intricate similarity structure cues to infer spatial arrangements. However, the similarity structure alone does not explicitly define which group or

PLOS Computational Biology

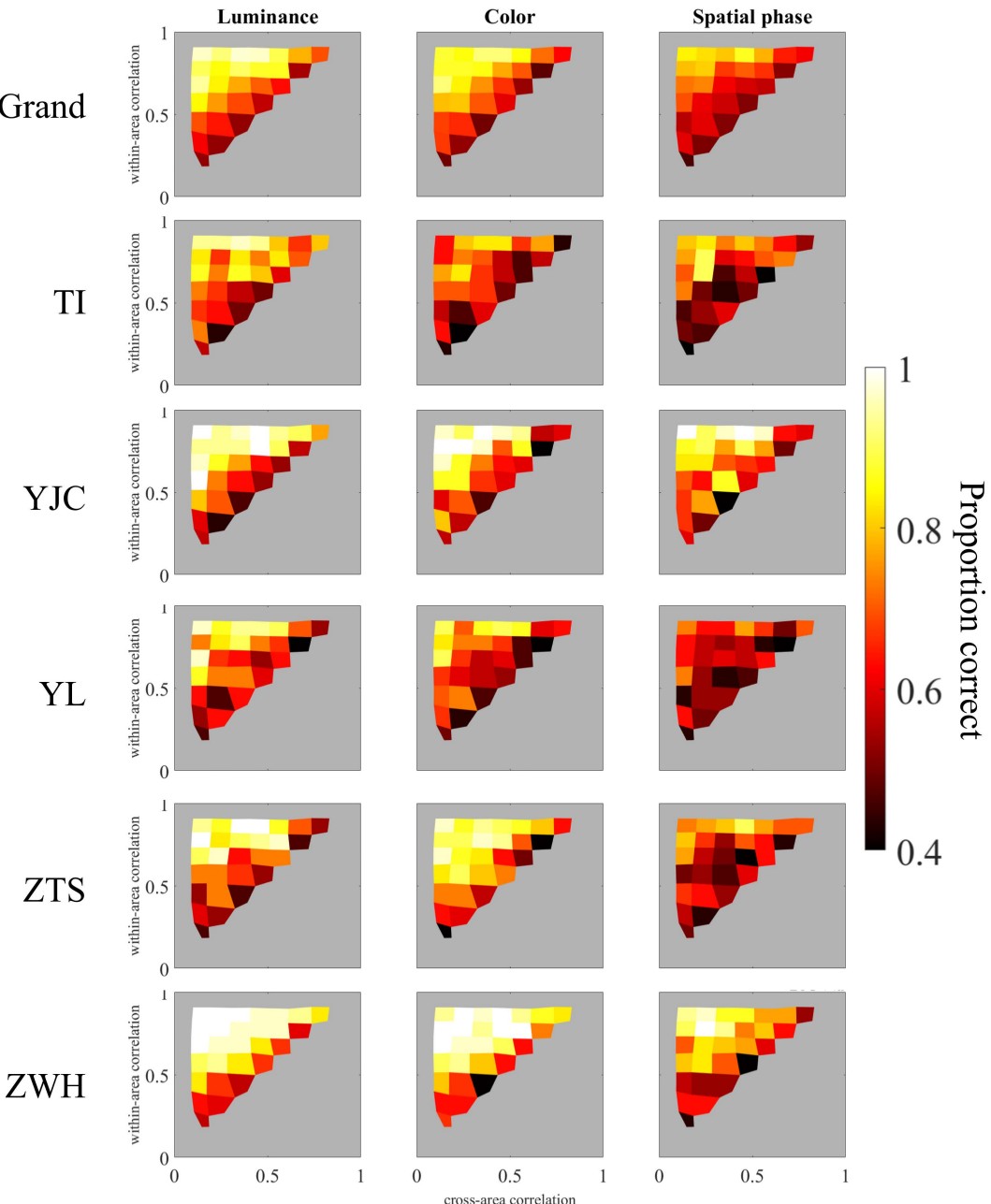

**Fig 5. Perceptual segmentation results based on luminance, color, and spatial phase.** The first row presents the grand average results across five participants, while individual participant data are shown in subsequent rows. Columns represent the results for the luminance, color, and spatial phase conditions. Within each graph, the x-axis represents the empirical cross-area temporal correlation of the generated stimuli, while the y-axis represents the empirical within-area temporal correlation. The proportion of correct responses is represented by reddish intensity, with contour lines included. The gray regions outside the triangular area indicate untested conditions, while conditions within the triangular area have values < 0.4.

object an individual pixel should be assigned to. Consequently, the visual system must employ an algorithm to convert similarity structure cues into predictions regarding object or group assignments without supervision, meaning it does not rely on predefined correct answers.

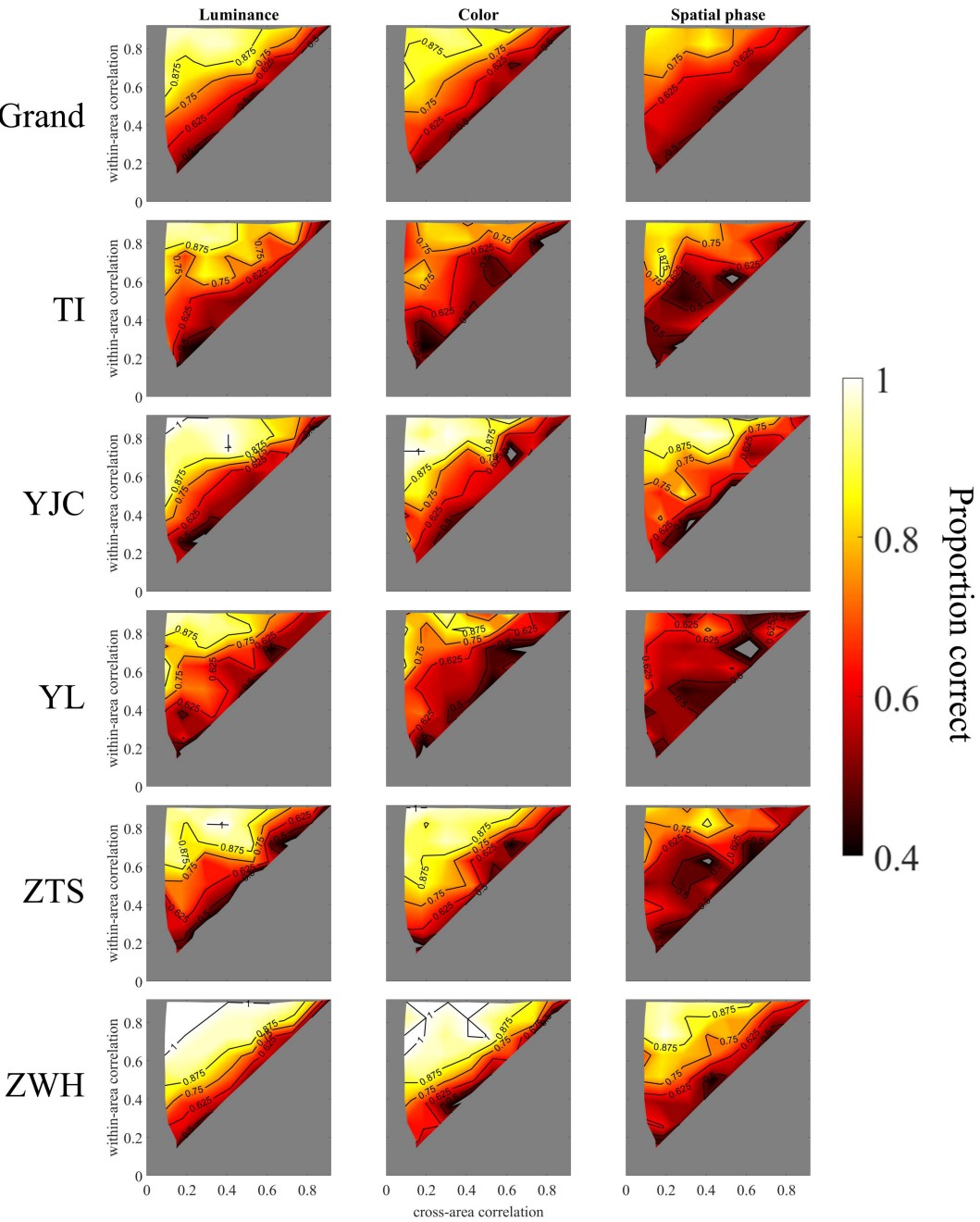

**Fig 6. The contour plot version upon Fig 5 to show the general tendency of the performance.**

To address this challenge, we utilized a classical graph cut model to simulate segmentation performance for our stimuli and assessed the extent to which the model can replicate human performance. Fig 7 illustrates the simulation protocol and model architecture. Similar to human participants, the model processed two video clips, a signal interval, and a null interval, and determined which clip can be divided into left and right spatial partitions. The model underwent multiple trials, and segmentation accuracy was measured as the proportion of correctly identified signal intervals. The proportion correct was computed across all 45 temporal correlation pairs tested in human experiments.

(A)

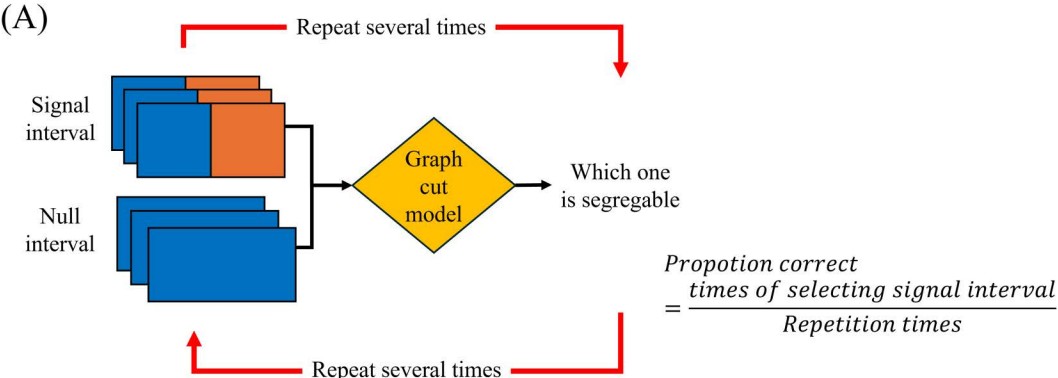

(B)

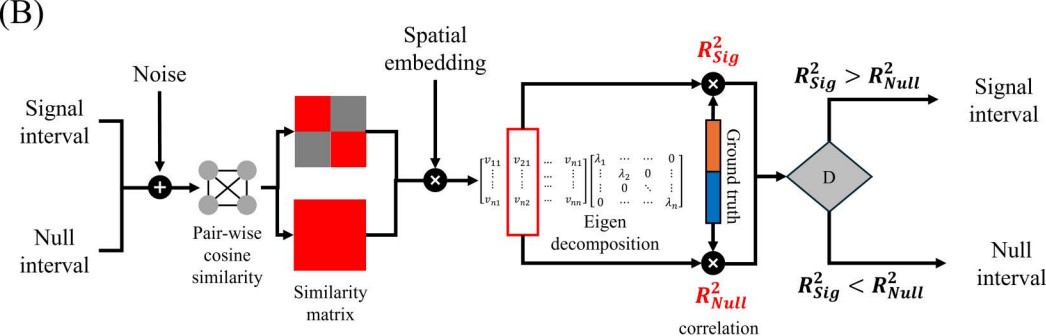

**Fig 7. (A)** Simulation protocol for measuring model visual segmentation performance. **(B)** Graph cut model architecture.

There were two model architecture versions: a naïve model and a full model. The naïve model did not include any trainable parameters and followed a three-stage processing sequence to evaluate the inherent capabilities of the existing structure. First, the model computed a pairwise temporal cosine similarity matrix, reflecting the experimental finding that human visual processing relies on similarity structures to infer spatial configurations. Next, it performed eigendecomposition on the similarity matrix to generate a spatial segmentation representation, thereby identifying group memberships for each pixel in an unsupervised manner. Finally, the model compared the estimated group memberships with the true ones and selected the interval (signal or null) maximizing similarity. This naïve model represents the best-case scenario under no internal limitations and serves to reveal the full computational capacity of this graph-based approach.

In the full model, we introduced two adjustable parameters (Fig 7) to refine the model's performance and better align it with human capabilities. The first parameter incorporated noise before computing the similarity matrix, with the signal-to-noise ratio (SNR) controlling the noise level. Noise is an inherent aspect of human visual processing and disrupts similarity computations, leading to variability in perceived similarity strength. In our model, both noise addition and similarity computation were linear operations, meaning that the order of these steps—whether noise is introduced before or after similarity computation—did not affect the final output.

The second parameter accounted for the possibility that the visual system may not evaluate all similarity pairs equally. Previous studies [34,35] have suggested that the ability to judge simultaneity declines as the spatial distance between elements increases, potentially due to longer signal transmission times or reduced neural connectivity between distant spatial locations. To replicate this effect, we introduced a distance-dependent weighting function in the similarity matrix, reducing the contribution of similarity strength for elements that are farther apart. The position-embedding sigma controlled this weighting as a function of distance. A higher sigma value allowed the model to consider more widely separated element

pairs, whereas a lower sigma value restricted similarity computations primarily to neighboring elements, such as those at the segmentation boundary.

## Methodology

### Model architecture

This section presents the model architecture tested in the simulation. The model utilized temporal correlation to construct a fully connected undirected graph across spatial locations, finding the coarsest subgraph as the estimation of segmentation.

The model input stimuli of the same size as those generated by the ViT ($8 \times 16 \times 30$). Two sequences, signal interval ($F^*_{sig}$) and null interval ($F^*_{Null}$), were simultaneously used as model input. In most cases, the calculations applied to both the signal and null intervals. Unless otherwise specified, only the computations for signal interval are presented, with the null interval being processed in the same manner.

After the stimuli were input, we added internal noise to the input as follows:

$$F'_{sig} = \frac{\tau}{1+\tau} \cdot F^*_{sig} + \frac{1}{1+\tau} \cdot \mathcal{E} \tag{16}$$

where $\mathcal{E}$ is Gaussian white noise with the shape $8 \times 16 \times 30$, a mean of 0, and variance of 1. $\tau$ represents the free parameters controlling the SNR within the range of $[0, \infty)$.

After adding noise, we flattened $F'_{sig}$ from 3D to 2D, resulting in $X_{Sig}$ with dimensions $(8 \times 16) \times 30$. The mapping relationship was as follows:

$$F'_{Sig}(k, t) = X_{Sig}\left(\left\lceil \frac{k}{8} \right\rceil, mod(k, 8), t\right)$$
$$k \in \{1, 2, \ldots, 96\} \tag{17}$$

where $\lceil \cdot \rceil$ is the ceiling function, returning the smallest integer greater than or equal to the argument, and where $mod(\cdot)$ returns the remainder after division.

Next, we calculated the similarity matrix based on $X_{Sig}$. The cosine similarity between any pair of vectors $i$ and $j$ was given by:

$$S_{Sig}(i,j) = \left| \frac{X_{Sig}(i,:)X_{Sig}(j,:)^\top}{\|X_{Sig}(i,:)\|_2 \|X_{Sig}(j,:)\|_2} \right| \cdot \Omega(i,j), \; \forall i, j \in \{1, 2, \ldots, 96\}$$
$$\Omega(i,j) = e^{-\frac{\Delta(i,j)^2}{\sigma_\Delta^2}}$$
$$\Delta^2(i,j) = \left(\left\lceil \frac{i}{8} \right\rceil - \left\lceil \frac{j}{8} \right\rceil\right)^2 + (mod(i,8) - mod(j,8))^2 \tag{18}$$

Here, the similarity was computed as the absolute value of the cosine similarity, weighted by a Gaussian function $\Omega(i,j)$, which reduced the connectivity between distant elements. The parameter $\sigma_\Delta^2$ controlled the width of the Gaussian windows, with values in the range of $[0, \infty)$.

We then transformed the similarity matrix into an affinity matrix by applying a step function, turning it into a simple binary graph structure instead of a weighted graph:

$$A_{Sig}(i,j) = \begin{cases} 1, & S_{Sig}(i,j) \geq \overline{S_{Sig}} \\ 0, & otherwise \end{cases} \tag{19}$$

where $\overline{S_{Sig}}$ is the mean value of the whole similarity matrix.

Next, we performed a classical eigendecomposition on the normalized Laplacian matrix as follows:

$$L_{Sig} = I - D_{Sig}^{-\frac{1}{2}} A_{Sig} D_{Sig}^{-\frac{1}{2}}$$
$$D_{Sig} = diag\left(\sum_{j=1}^{8 \times 16} A_{Sig}(i,j)\right)$$
$$diag\left(V_{n \times 1}\right) = V_{n \times 1} \odot I_{n \times n} \tag{20}$$

The normalized Laplacian $L_{Sig}$ ensures all diagonal entries are 1, which equalizes the contribution of each node and stabilizes the matrix decomposition.

The eigendecomposition of the Laplacian matrix yielded:

$$L_{Sig} = v_{sig}^{-1} \lambda_{sig} v_{sig} \tag{21}$$

where $v_{Sig}$ represents the eigenvectors and $\lambda_{Sig}$ represents the eigenvalues. According to a previous study [36], the eigenvectors corresponding to the second smallest eigenvalue (also called the Fiedler vector) is the optimal solution for partitioning the graph into two subgraphs. We transformed the vector into matrix form as follows:

$$G_{Sig}(h,w) = v_{sig}\left((h-1) \times 8 + w, l\right), \tag{22}$$

where $l$ is the position corresponding to the second smallest eigenvalue. This process provided the coarsest graph structure. In simple binary segmentation tasks, such as figure-ground segmentation and in our task, the value of the Fiedler vector is directly proportional to the probability that a given spatial point belongs to the target.

Finally, we transformed the eigenvector into a decision-making process based on the correctness rate. Specifically, the model outputs the option with the highest similarity to the answer. The correctness rate was defined as follows:

$$R_{Sig}^2 = \left(\frac{\sum_h \sum_w (G_{sig}(h,w) - \overline{G_{sig}})(\mathbb{T}(h,w) - \overline{\mathbb{T}})}{\sqrt{\sum_h \sum_w (G_{sig}(h,w) - \overline{G_{sig}})^2}\sqrt{\sum_h \sum_w (\mathbb{T}(h,w) - \overline{\mathbb{T}})^2}}\right)^2$$
$$\mathbb{T}(h,w) = \begin{cases} 1, & 1 \le h \le 8, \ 1 \le w \le 8 \\ -1, & otherwise \end{cases} \tag{23}$$

where $\mathbb{T}(h,w)$ is the target matrix (i.e., the answer), $\overline{\mathbb{T}}$ is its mean, and $\overline{G_{Sig}}$ is the mean of the model estimation.

The model output the result based on the following decision rule:

$$D\left(R_{Sig}^2, R_{Null}^2\right) = \begin{cases} signal, & R_{sig}^2 > R_{Null}^2 \\ null, & R_{sig}^2 < R_{Null}^2 \\ \gamma, & otherwise \end{cases}$$
$$\gamma \sim Bernoulli(p = 0.5) \tag{24}$$

where $\gamma$ is a random variable following the Bernoulli distribution with equal probability of returning either signal or null.

## Simulation protocol

For each trial, the model generated two possible outcomes: signal or null. For each of 45 correlation pairs, this process was repeated 1,000 times, with input stimuli being randomly selected from 30 variants of a single temporal correlation pair and noise being regenerated for each iteration. Based on these 1,000 responses, we calculated the percentage of signal

responses, defining it as the proportion correct for a single correlation pair. We tested all 45 temporal correlation pairs to obtain the model's performance in the segmentation task, representing the proportion correct as a function of both within- and cross-area temporal correlation ($\Psi_{Model}$). The model's goodness of fit was determined by its ability to predict human responses ($\Psi_{Human}$), calculated as follows:

$$R^2 = 1 - \frac{\sum_{\phi_{cross}} \sum_{\phi_{within}} \left( \Psi_{Model}\left(\phi_{cross}, \phi_{within}\right) - \Psi_{Human}\left(\varphi_{cross}, \phi_{within}\right)\right)^2}{\sum_{\phi_{cross}} \sum_{\phi_{within}} \left( \Psi_{Human} - \overline{\Psi_{Human}}\right)^2}$$

(25)

where $\varphi_{cross}$, $\varphi_{within}$ represent cross- and within-area temporal correlation, respectively.

For the naïve model, the parameter was configured to have no impact on model performance, with the signal rate (i.e., $\frac{\tau}{1+\tau}$) set to 1, corresponding to an infinite SNR. Additionally, the position-embedding sigma (i.e., $\sigma_\Delta^2$) was set to infinity, causing the weight function to become 1 universally. To determine the most suitable parameter pair for replicating the human result, we implemented a grid search sampling approach in the parameter space. We selected 50 samples with linear intervals from $0 \leq \frac{\tau}{1+\tau} \leq 1$ for the SNR. Similarly, for the position-embedding sigma, we chose 50 samples with linear steps from $0 \leq \sigma_\Delta \leq 16.5$, where the maximum value represents the largest Euclidean distance in our stimuli. This resulted in 2,500 parameter pairs being sampled across the parameter space, which were then used to assess the model's goodness of fit against human responses.

## Results

### Naïve model simulation results

Fig 8 presents the simulation results of the naïve model, showing the optimal performance of the graph-cut algorithm without biologically plausible constraints included in the full model. The top section of the figure displays a single trial output from the second stage. Without data supervision, the model's estimated group memberships form clear boundaries that largely align with the true memberships, except in the null interval (other examples of output can be found in the supplementary material). In terms of segmentation performance, the model's outcomes vary based on both within- and cross-area temporal correlations, similar to human performance. However, unlike humans, the model achieves near-perfect segmentation even when the difference between within- and cross-area temporal correlations is minimal.

### Full model simulation results

Fig 9 presents the effects of the two parameters on the model's output, while F presents a 2D predictability heatmap based on these parameters, along with the optimal model for three attributes. The analysis reveals that SNR is the primary factor influencing the model's overall performance, whereas the sigma parameter has a minor impact. When the position-embedding sigma exceeds a certain threshold (e.g., greater than one element), predictability remains nearly constant across attributes, regardless of further changes in sigma.

The optimal model (see Fig 10) achieved high predictability levels for the three attributes, with respective $R^2$ values of 0.96, 0.93, and 0.90. The corresponding SNR values were 0.48, 0.48, and 0.4, while the spatial embedding sigma values were 7.09, 4.73, and 7.09. These findings suggested that with appropriate parameter settings, the model can achieve performance comparable to human segmentation abilities. The lower performance in the spatial phase can be attributed to a lower SNR compared to luminance and color.

Furthermore, the position-embedding sigma primarily influenced the slope of the contour, as shown in the third row of Fig 8. Higher sigma values resulted in a slope closer to the diagonal line (i.e., within-area temporal correlation = cross-area temporal correlation). In contrast, when sigma was small, the model prioritized only short-distance temporal correlations, causing within-area and cross-area temporal correlations to contribute asymmetrically to segmentation performance.

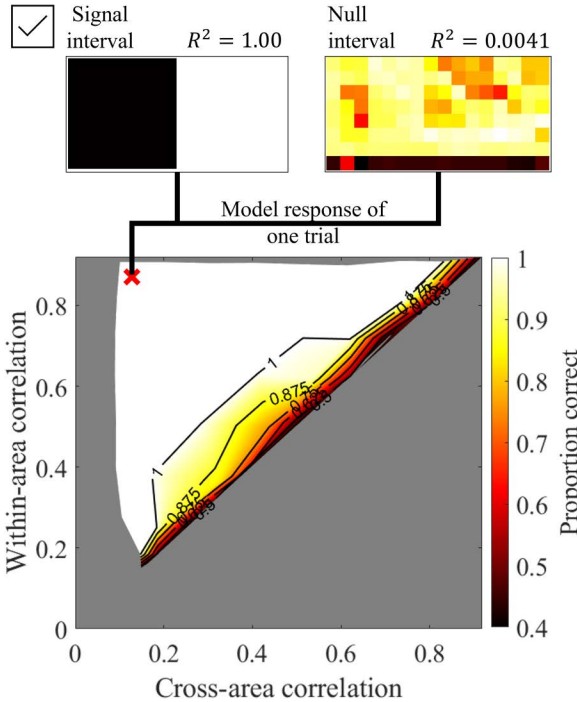

**Fig 8. Simulation result for the naïve model.** The one-trial demo output for the second stage of the naive model is shown at the top of the figure, with the signal interval at the left and the null interval at the right (other examples of output can be found in the supplementary material). The temporal correlation pair shown is labeled as a red cross symbol on the proportion correct heatmap.

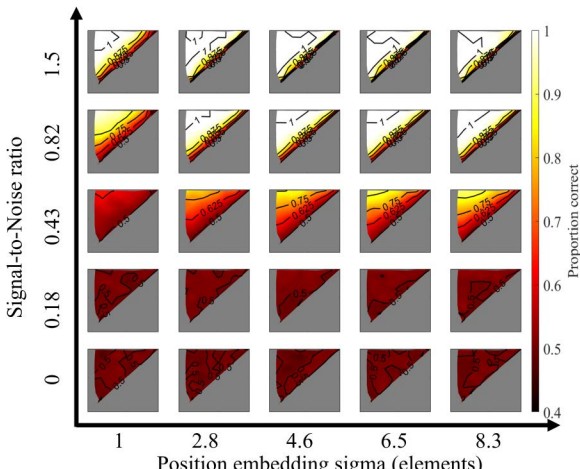

**Fig 9. Model output of different pairs of parameters.**

The model's predictability in simulating human performance suggested that the position-embedding sigma cannot be too small, indicating that similarity across certain distant elements also plays a crucial role. This finding indicated that human visual processing incorporates not only short-range similarities, such as those between neighboring elements, but also long-range similarities across spatially distant elements.

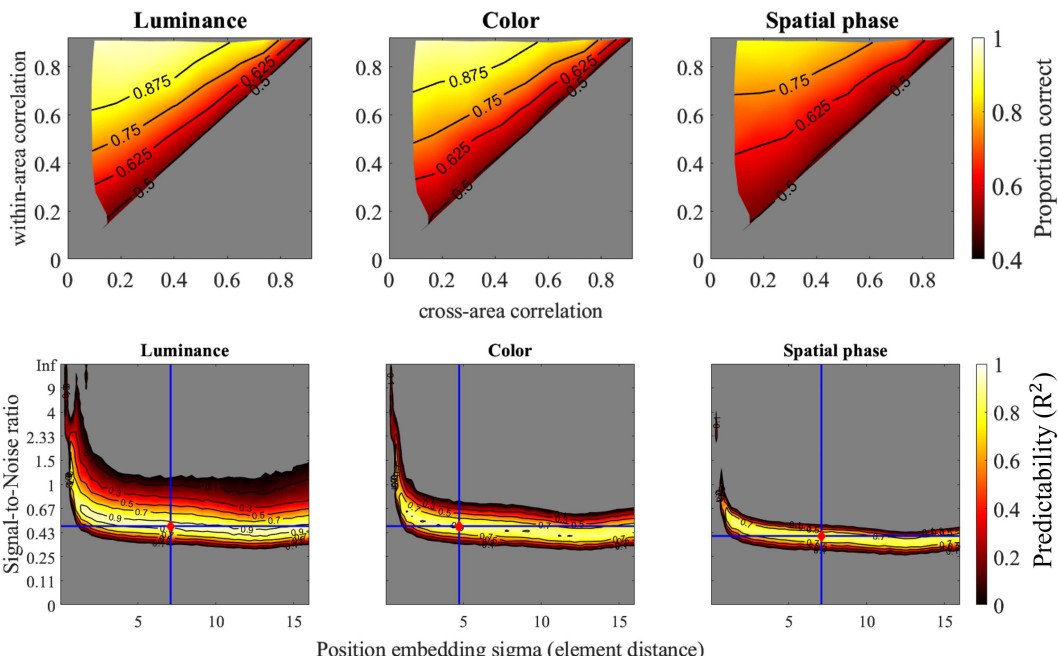

**Fig 10. Simulation of dense parameter pairs and optimal model output.** The upper section of the figure displays the model's simulated proportion-correct values under the optimal parameter pair. The lower section presents a heatmap illustrating how well the model predicts human responses to different attributes. The optimal parameter pairs are marked as red dots, with accompanying blue vertical and horizontal lines indicating their locations on the graph.

## General Discussion

This study used ViTs to generate stimuli with precisely controlled within- and cross-area temporal correlations, allowing us to examine how the human visual system integrates similarity structures for segmentation. Our findings indicated that segmentation is not solely dependent on local comparisons between individual elements, but also involves a global process that integrates similarity across regions. Furthermore, we demonstrated that a graph-based computational model effectively captures human segmentation behavior, underscoring the potential role of structured relational processing in mid-level vision.

## Transformer-based generation of complex stimuli

A key contribution of our work is the introduction of a novel protocol for generating temporally complex stimuli with precisely controlled element-wise temporal synchrony. Previous studies investigating temporal synchrony have primarily relied on simple periodic waveforms, such as sinusoidal waves [15,35] or square waves [11,14,37]. While some studies have explored non-periodic waveforms by randomly assigning temporal shifts to individual elements [13,23], these methods generally lack precise control over synchrony between individual elements.

The primary challenge in modulating element-wise temporal synchrony within a stochastic time series lies in constructing a covariance matrix that enforces a specific similarity structure. When the number of elements is large, the covariance matrix often becomes singular, making it impossible to directly sample from a statistical distribution, such as a multivariate normal distribution, while maintaining the desired temporal correlation constraints. One potential solution is an iterative approach in which stimuli are randomly generated and tested against the desired similarity structure. However, this method is computationally expensive and inefficient, requiring an unpredictable amount of time to produce a sufficient number of trials.

In this study, we addressed these limitations by leveraging a ViT-based approach to directly optimize the temporal similarity structure. By defining a similarity matrix as a loss function, we iteratively refined the stimuli to precisely match the desired within-region and cross-region temporal correlation constraints. This method not only ensures that the generated stimuli adhere to strict statistical properties but also provides a scalable solution for designing complex temporal textures. Our protocol offers a more flexible and computationally feasible approach for future research on spatiotemporal grouping, temporal synchrony, and other mid-level vision phenomena.

In this work, we used a ViT-based generation model to generate stimuli for a psychology experiment. However, other large-parameter models could also achieve similar results. In deep learning, this approach can be understood as a form of over-parameterized optimization. The solution space defined by the loss function over the original stimulus is usually non-convex, making direct optimization in the original parameter space difficult. By introducing a model with more parameters and stronger representational capacity, we effectively map the problem into a higher-dimensional space. This reduces the risk of getting stuck in local minima and makes it easier to find stimuli that meet the experimental goals. ViT is simply the architecture we chose for this study, but in principle, other sufficiently powerful models could also be used.

Worthy to note, the stimuli used in this study followed a fixed and simple geometric layout with a clear segmentation border. Combined with a strict 2IFC method, the segmentation thresholds we estimated might be too low to characterize region segregation performance under natural scenes where segregation borders could be more complex and uncertain. Such naturalistic segmentation performance may be effectively quantified using the method proposed by [38]. In addition, because this method estimates region segregation maps by measuring pairwise similarities between numerous local points, combining it with our stimuli enables a direct evaluation of how accurately human observers judge pairwise temporal correlations between elements in a scene. This extension may also help to clarify how strong does the segmentation relies on local or global similarity cues, and how boundary location is inferred from global temporal structure.

Another limitation of the present study relates to the relatively small number of participants. Such a small sample cannot fully capture the population variance in the ability of temporal segmentation. The current study focused instead on within-participant variance to obtain accurate estimates at the individual level. Future studies with larger and more diverse participant groups will be important to better capture general tendencies and characterize response variability.

## Graph-based representation and its role in mid-level vision

In our experiment, participants exhibited better segmentation performance when within-region temporal correlation was high and cross-region temporal correlation was low. This suggested that the visual system does not rely solely on individual local features, but instead integrates similarity information across multiple spatial points. Given that low-level cues (e.g., luminance, contrast, and spatial frequency) and high-level semantic information were unavailable, the mid-level visual system must rely on the relative temporal similarity between elements to infer the segmentation structure. This finding aligned with previous studies indicating that mid-level vision constructs representations based on relational properties rather than absolute local features, such as non-accidental properties [39], configural relationships (e.g., T- and L-junctions) [40], surface inference mechanisms [41,42], Wallis et al. [43] suggest that segmentation and grouping mechanisms may be mediated by both local interactions between nearby image features and global properties of the scene. Our study suggests that the same may hold true for grouping and segmentation based on temporal correlation.

A natural way to formalize such relational structures is through a graph-based representation. In this framework, individual elements in the stimulus can be conceptualized as nodes, while the temporal similarity between them determines the edge weights. The segmentation process then resembles graph partitioning, where regions are separated based on internal consistency and cross-boundary dissimilarity. Various methods exist for implementing graph partitioning; here, the eigendecomposition we employed is a simple linear transformation of the graph structure to extract the second-largest principal component for this task. This approach was based on previous studies demonstrating that the second-largest principal component represents the coarsest structure derivable from a graph [18,36]. Additionally, computational

neuroscience research suggests that principal component analysis can be efficiently implemented with a small number of neurons, supporting the physiological feasibility of such computations in the brain [44]. In our simulation, we observed that graph partitioning based on the second-largest principal component effectively replicates human segmentation performance with minimal parameterization.

Our model simulations also suggested that differences in segmentation performance across visual attributes may arise from variations in SNR. In particular, the lower segmentation performance for spatial phase indicates that higher spatial frequency signals require stronger contrast and lower temporal frequency to achieve detection levels comparable to those of low spatial frequency stimuli [45,46]. This SNR difference also aligns with our experimental design. Because the same set of 30 pre-generated temporal tensors was used across all three attributes, and spatial differences such as mean luminance and contrast were either matched or negligible, the physical-level differences between attributes were minimized. Therefore, the observed SNR differences most likely reflect intrinsic signal gain or tuning bandwidth differences across the processing channels for the three attributes.

Moreover, our results suggested that restricting comparisons strictly to neighboring elements, as controlled by the position-embedding parameter, does not fully account for human segmentation performance. Instead, long-range comparisons are necessary, aligning with previous studies demonstrating long-range temporal comparison mechanisms in motion perception [34]. However, interactions between temporal and spatial information are complex [35,47] and warrant further investigation.

Alternative computational approaches could also yield results comparable to our graph-based model. For instance, while image statistics are represented in the primate cortex [37], image segmentation could be achieved by decomposing feature distributions through probabilistic inference [4,38]. Future research should focus on identifying key stimuli and experimental paradigms to determine which algorithms are actually employed by the human visual system.

## Comparison with CV image segmentation models

While image segmentation models in the field of CV effectively utilize spatial similarity structures for segmentation, they largely disregard the temporal aspect of the task. As a result, these models struggle with segmentation based on temporal similarity. This limitation arises because most CV models process images in a frame-by-frame manner, often neglecting the global covariation of local pixels over time. In other words, current CV approaches lack mechanisms for integrating global temporal information. In contrast, human vision relies on a global process that integrates temporal covariation signals to infer spatial structures [11,13,14,24].

At present, no mature CV models fully address segmentation based on temporal similarity. However, our study leverages generated stimuli to investigate human perception of temporal similarity and applies a graph cut algorithm to simulate a global process that selectively integrates pairwise similarities. This approach better approximates human performance compared to conventional CV models. These findings suggest that incorporating global temporal similarity processing into CV models may be a promising avenue for advancing dynamic image segmentation.

## Relationship with current neurophysiological evidence

Neurophysiological research suggests that visual segmentation involves multiple brain regions, with specific areas recruited depending on stimulus complexity and feature composition. Numerous studies have indicated that recurrent signaling between V1, V2, V4, and the inferior temporal cortex plays a crucial role in contour integration and simple shape perception [48–50].

For segmentation based on static visual features such as luminance, color similarity, or spatial proximity, functional magnetic resonance imaging and event-related potential studies have shown that contour-based segmentation primarily activates the lateral occipital cortex in the ventral pathway [51,52]. However, when segmentation involves more complex closed boundary shapes or object processing, activity shifts toward V3 and the intraparietal sulcus in the dorsal pathway

[53–55]. The parietal lobe, particularly the inferior intraparietal sulcus, has been implicated in integrating different feature representations and processing hierarchical relationships within the visual scene [49]. In contrast, segmentation based on dynamic features, such as structure-from-motion, engages distinct regions, with the medial temporal area playing a central role [56].

Currently, no definitive neuronal evidence identifies the exact brain regions responsible for visual grouping or segmentation based on temporal similarity. Some studies suggest that the inferior parietal lobe and the insula contribute to processing temporal relationships, particularly temporal synchrony and asynchrony [57–59]. This raises the possibility that temporal similarity-based segmentation relies on mechanisms distinct from those governing segmentation based on static or motion cues.

The neural basis of graph-based computation in segmentation remains an open question. Future research should explore whether the brain employs an explicit graph-based representation for visual segmentation and, if so, which regions support this computation. Given their roles in relational processing and feature binding, the intraparietal sulcus and inferior parietal lobe are potential candidates for encoding similarity structures in a graph-like manner. Integrating neuroimaging techniques with computational modeling may help elucidate the neural mechanisms underlying mid-level visual segmentation.

## Conclusion

Summing up, this study leveraged a Vision Transformer–based framework to generate stimuli with precisely controlled temporal similarity structures, enabling a systematic investigation of how the human visual system infers spatial organization from complex temporal correlations among elements. The results suggest that human segmentation behavior strongly depends on parsing global similarity structures to derive near-optimal grouping solutions, rather than relying solely on local comparisons.

This finding implies that to further understand how the visual system resolves spatial structure, it is necessary to consider more structured computational models and identify stimulus designs that can probe these mechanisms more effectively.

### Resource availability

All resources, including the experiment code, demo video, raw data, high-resolution figures in the paper, and bootstrapping results, have been uploaded to the Open Science Framework platform. The project can be accessed at the following link: https://osf.io/9yncd/?view_only=eb481f2b28ab42e29ff267833f70b8e5.

### Supporting information

**S1 Appendix. Demo output with the signal interval input.**
(DOCX)

**S2 Appendix. Demo output with the null interval input.**
(DOCX)

### Author contributions

**Conceptualization:** Yen-Ju Chen.

**Data curation:** Yen-Ju Chen.

**Formal analysis:** Yen-Ju Chen.

**Funding acquisition:** Shin'ya Nishida.

**Investigation:** Yen-Ju Chen.

**Methodology:** Yen-Ju Chen, Zitang Sun.

**Project administration:** Shin'ya Nishida.

**Resources:** Shin'ya Nishida.

**Supervision:** Shin'ya Nishida.

**Visualization:** Yen-Ju Chen.

**Writing – original draft:** Yen-Ju Chen, Zitang Sun.

**Writing – review & editing:** Zitang Sun, Shin'ya Nishida.

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
