## [Decision Letter · Decision Letter 0]

6 Jul 2025

PCOMPBIOL-D-25-00570

Human visual grouping based on within- and cross-area temporal correlations

PLOS Computational Biology

Dear Dr. Chen,

Thank you for submitting your manuscript to PLOS Computational Biology. After careful consideration, we feel that it has merit but does not fully meet PLOS Computational Biology's publication criteria as it currently stands. Therefore, we invite you to submit a revised version of the manuscript that addresses the points raised during the review process.

Please submit your revised manuscript within 60 days Sep 05 2025 11:59PM. If you will need more time than this to complete your revisions, please reply to this message or contact the journal office at ploscompbiol@plos.org. Please include the following items when submitting your revised manuscript:

We look forward to receiving your revised manuscript.

Kind regards,

Yuanning Li

Academic Editor

PLOS Computational Biology

Daniele Marinazzo

Section Editor

PLOS Computational Biology

**Journal Requirements:**

4) Please amend your detailed Financial Disclosure statement. This is published with the article. It must therefore be completed in full sentences and contain the exact wording you wish to be published.

2) If any authors received a salary from any of your funders, please state which authors and which funders..

5) The file inventory includes files for Figures 5a and 5b. We would recommend either combining these into a single Figure 1.tiff file with separate internal panels, or renumbering them as individual figures, as we are not able to publish multiple components of a single figure as separate files.

**Reviewers' comments:**

Reviewer's Responses to Questions

**Comments to the Authors:**

Reviewer #1: This study leverages vision transformer models to create stimuli where the temporal correlation within and between regions of a stimulus can be robustly manipulated. They first explain the details of this model’s implementation and then test humans’ ability to distinguish between images with two distinct subregions and images without distinct subregions. They find that participants are sensitive to global similarity relations and do not merely depend on pairwise similarity comparisons between two elements. Finally, they develop a computational model that, when perceptual noise is included as a parameter and fit to human performance, predicts human performance on the psychophysical task quite accurately.

I liked this paper and think that it makes good contributions to the study of midlevel vision. I have a few suggestions that I would like to see addressed before I recommend it for publication.

First, and most importantly, I think the paper would be improved by making the questions addressed in Studies 2 and 3 of the study earlier and more clearly. In particular, after Study 1 (p. 15), the authors go very quickly into experimental parameters for Study 2. This reader would have been helped by a more thorough explanation of what was being tested and what the relevant hypotheses were going into Study 2.

For Study 3, I was a little unclear about whether the naïve model was the best foil to compare with the parameterized model. It seems predictable that a model that directly computes cosine similarity will perform near-optimally and make few humanlike errors. Under what hypothesis would human performance match the naïve model better or equally well to the parameterized model? Is there another model, potentially a non-graph cut model whose differences in human predictability would be more illuminating?

Finally, I think the authors could make clearer what was learned about human perception in the General Discussion. The GD does have some of this, and I like what is written, but given the breadth of readership at PLOS CB, I think many readers would benefit from a little additional explanation of the key takeaways.

Minor:

On page 6 (first bullet point), the authors say that W is short for “weight”. Corrrect me if I’m wrong, but should this be “width”?

Reviewer #2: In this manuscript, the authors tested an interesting research question: how does the human visual system segment image areas using spatial and temporal similarities. To better answer this question, the authors introduces a three-stage programme aimed at understanding how global temporal-similarity structures guide visual segmentation. (1) A Vision-Transformer-based generator (Exp 1) creates dynamic 8 × 16 noise images whose only diagnostic cue is a user-specified matrix of within-area and cross-area cosine correlations. (2) In a 2IFC psychophysical task (Exp 2) observers decide which of two 1s video clips contains a segmentable texture (created in Exp1); performance grows monotonically with the difference between the 'within group similarity' and 'between group similarity'. (3) A computer vision model (Exp 3) that capture the human performance and illustrate the detialled computational mechanisms of human visual texture segmentation. The authors conclude that the human visual system performs a near-global, similarity-weighted computation that can be approximated by a noisy spectral-clustering algorithm.

The work is timely, technically ambitious and, in principle, sound. The stimulus generator is elegant (but some justification is needed), the behavioural gradients are clear (although there are some limitaitons), and the modelling framework offers a quantitative bridge between vision science and computer vision methods. However, several inconsistencies in the methods section, together with unanswered questions about statistical treatment and parameter sensitivity, currently limit interpretability. Addressing the issues below will strengthen both the empirical foundation and the theoretical claims.

Major Concerns:

1. The stimuli generation methods (Exp1). I found a lot of details are missing. First, I think I understand that each frame is in-distinguishable between areas, the temporal information allows such distinguish, am I right? However, I am a bit worried by the consistant and clear cut off between the two areas (easy to calculate).

For instance, a recent paper (Vacher, J., Launay, C., Mamassian, P., & Coen-Cagli, R. (2023). Measuring uncertainty in human visual segmentation. PLoS Computational Biology, 19(9), e1011483.) tested a similar research question, but do not have such a rigorous and pre-defined boundary (or box).

2. The human data collection needs more details. (1) How many subjects? The authors said there are 4 (2 authors + 2 naive subjects), however, there are 5 participants' data in Figure 5 (excluding the 'general' one). I think this is very concerning. (2) Details of the procedure. This Expiement 2 is very long (7.5 hrs), pls specify the duration of each block and the gap/rest/interval between for future replication. (3) The major concern lies in here: if half of the subjects are the authors, even if they cant figure out the 'correct answer' simply from images (nice image generation from Exp 1), they are still aware the stimuli were seperated at the centre. SO it is very possible for them (which consistute half of the data!) to simply rely on the centre part of the videos/stimuli (not necessarily the whole image) to make the judgement. Yes, the stimuli are so 'uniform' in this away.

Moreover, the design of Exp 2 is a bit off, why 2IFC? Is it possible that both videos are very much seperable but they have to chose one, or vice versa? Some additional details are necessary.

3. The Exp3's data suggests the SNR is an important factor, see minor concern for more details. I am a bit worried (as the authors stated in discussion), even this model could not read the 'temporal' aspect and so the stimuli are flatted from 3D to 2D. Does this affect the interpretation of the data?

Minor concerns:

1. Behavioural data suggests the seperation ratios are “Luminance > Colour > Phase”.

The difference is compatible with known channel SNR differences (as mentioned in Exp3), yet the manuscript currently leaves the reader wondering if uncontrolled cues leaked through stimulus generation. A discussion or justification is necesary.

2. Does one need the ViT to create the stimuli? Is it possible we can create these stimuli with a rigorous pre-defined algorithm? Does the ViT neccesary here?

Reviewer #3: 1. Originality

Up to my knowledge no previous works have been focusing on temporal cues of spatial segmentation

2. Innovation

The work propose a method to generate well controlled stimuli to conduct a precise study about visual segmentation

I've spent some time trying to generate these stimuli without the proposed method to convince myself that the method is necessary to generate appropriate stimuli.

3. High importance to researchers in the field

The results are quite remarkable and relevant for the studies around perceptual organization

4. Significant biological and/or methodological insight

Standard psychometric methods

5. Rigorous methodology

Yes

6. Substantial evidence for its conclusions

Yes

What are the main claims of the paper and how significant are they for the discipline?

The mains claims are :

- Humans are able to successfully segment images based on the spatial integration of temporal correlation cues.

- The data can be explained by a graph-based model that integrate pairwise correlations across the image

Are these claims novel? If not, which published articles weaken the claims of originality of this one?

I am not aware of other sharp results like this regarding visual segmentation.

Are the claims properly placed in the context of the previous literature? Have the authors treated the literature fairly?

Yes

Do the data and analyses fully support the claims? If not, what other evidence is required?

Yes

Would additional work improve the paper? How much better would the paper be if this work were performed and how difficult would it be to do this work?

In many places the authors talk about similarity or correlation, they must precise that it is temporal (or not) ! (eg l461)

Section Design and procedure (l383) : the writing must be improved/simplified. I had the impression to read the same info twice written in slightly different manner.

The section contain redoundancies :

- "introduce three different attributes" / "we examine three attributes"

- "viewed two consecutive 1-second video clips" / "view 1.000 ms video clips"

The number of repetitions for each of the 45 cor pairs is not indicated.

Section Model architecture (l487) : Isn't it F'_sig that is used to compute S_sig ??

Typos :

l51 : computational model

l307 : missing ref for figure

l374 : it seems you have 5 participants not 4

l513 : it's unclear what is the normalized Laplacian matrix

l527 : \bar T is the target not T(h,w)

Questions :

- Eq 9 why absolute values instead of squared ? Eq 11 why norm-1 is used ? Isn't it a regular euclidean norm ? (in fact he question hold for all norms !)

- Figure 5 : why the checkerboards are not straight ? It is unclear what these weird shapes are representing.

Complementary references for discussion

- Vacher, J., Launay, C., Mamassian, P., & Coen-Cagli, R. (2023). Measuring uncertainty in human visual segmentation. PLoS Computational Biology, 19(9), e1011483

- Wallis, Thomas SA, et al. "Image content is more important than Bouma’s Law for scene metamers." ELife 8 (2019): e42512.

Has the author-generated code that underpins the findings been made publicly available?

Yes

Are details of the methodology sufficient to allow the experiments to be reproduced?

With the minor corrections yes

Is the manuscript well organized and written clearly enough to be accessible to non-specialists?

Overall yes

Does the paper use standardized scientific nomenclature and abbreviations? If not, are these explained at the first usage?

Yes

**Have the authors made all data and (if applicable) computational code underlying the findings in their manuscript fully available?**

Reviewer #1: Yes

Reviewer #2: Yes

Reviewer #3: Yes

**Figure resubmission:**
---

## [Decision Letter · Decision Letter 1]

22 Aug 2025

PCOMPBIOL-D-25-00570R1

Human visual grouping based on within- and cross-area temporal correlations

PLOS Computational Biology

Dear Dr. Chen,

Thank you for submitting your manuscript to PLOS Computational Biology. After careful consideration, we feel that it has merit but does not fully meet PLOS Computational Biology's publication criteria as it currently stands. Therefore, we invite you to submit a revised version of the manuscript that addresses the points raised during the review process.

Please submit your revised manuscript within 30 days Oct 22 2025 11:59PM. If you will need more time than this to complete your revisions, please reply to this message or contact the journal office at ploscompbiol@plos.org. Please include the following items when submitting your revised manuscript:

We look forward to receiving your revised manuscript.

Kind regards,

Yuanning Li

Academic Editor

PLOS Computational Biology

Daniele Marinazzo

Section Editor

PLOS Computational Biology

**Additional Editor Comments:**

The reviewers are in general happy with the revisions, with only a few minor points that need to be addressed before final acceptance. Please consider these comments in a final round of revision.

**Reviewers' comments:**

Reviewer's Responses to Questions

**Comments to the Authors:**

Reviewer #1: I thank the reviewers for addressing my concerns. I think the paper is suitable for publication apart from a few typos.

p. 7: "In our implementation, we used several

188 four stacked ResidualBlock2D modules, each consisting of convolution, normalization

189 (InstanceNorm2D), and GELU activation, with the residual connection between two layers." Should be corrected

p. 17: As reviewer 2 pointed out, the demographic information is not complete. There were 5 participants, but only four participants' demographic information was reported.

Reviewer #2: In this revision, the authors followed the comments and made the manuscript into a better shape. They updated some very important missing information in the manuscript. The updated neurophysiology part is good. Also, the updated discussion section strongly extends the readability and clarity of the manuscript.

I found the updated general discussion and the method section very good. I think most of my concerns are solved. But I still want to point out that in the current human visual cognition/visual perception studies, we tend to have a larger sample size. While I believe the authors carried over some tendency from animal (primates) studies (like testing a lot of trials on two monkeys), I still suggest that they carefully note this limitation in the discussion.

Reviewer #3: I thank the authors for their responses and I have no remaining remarks. I think the paper is ready for publication.

**Have the authors made all data and (if applicable) computational code underlying the findings in their manuscript fully available?**

Reviewer #1: Yes

Reviewer #2: None

Reviewer #3: Yes

PLOS authors have the option to publish the peer review history of their article (what does this mean? ). If published, this will include your full peer review and any attached files.

**Do you want your identity to be public for this peer review?** For information about this choice, including consent withdrawal, please see our Privacy Policy .

Reviewer #1: No

Reviewer #2: No

Reviewer #3: No

**Figure resubmission:**
---

## [Editor Report · Decision Letter 2]

30 Aug 2025

Dear Dr. Chen,

We are pleased to inform you that your manuscript 'Human visual grouping based on within- and cross-area temporal correlations' has been provisionally accepted for publication in PLOS Computational Biology.

Best regards,

Yuanning Li

Academic Editor

PLOS Computational Biology

Daniele Marinazzo

Section Editor

PLOS Computational Biology

---

## [Editor Report · Acceptance letter]

PCOMPBIOL-D-25-00570R2

Human visual grouping based on within- and cross-area temporal correlations

Dear Dr Chen,

I am pleased to inform you that your manuscript has been formally accepted for publication in PLOS Computational Biology. Your manuscript is now with our production department and you will be notified of the publication date in due course.

With kind regards,

Zsofia Freund
